# NAIL: A CHALLENGING BENCHMARK FOR NAÏVE LOGICAL REASONING

## ABSTRACT

Logical reasoning over natural text is an important capability towards human level intelligence. Existing datasets are either limited and inadequate to train and evaluate logical reasoning capability (e.g., LogiQA and ReClor), or not oriented for logical reasoning (e.g., SQuAD and HotpotQA). In this paper, we focus on a specific category of logical reasoning, named *naïve logical reasoning*, and propose a new large-scale benchmark, named NAIL, aiming to help models learn and evaluate naïve logical reasoning capability. NAIL is sourced from standardized exams such as Chinese National Civil Servants Examination and Law School Admission Test. Furthermore, to collect more data, we propose to imitate examples of standardized exams rather than designing them from scratch. NAIL is available in both Chinese and English containing a total of $10,296 * 2$ instances. Empirical results show that state-of-the-art neural models struggle on NAIL with very poor accuracy (the best result is 30.1% for NAIL and 36.2% for Chinese NAIL), while human experts can perform 100% accuracy. Further results indicate that human imitations can significantly help models learn naïve logical reasoning ability.

## 1 INTRODUCTION

Current deep models have achieved near human-level performance on many tasks in NLP (Devlin et al., 2019; Liu et al., 2019), and more often than not, superficial knowledge suffices to solve the problems. To move towards human intelligence, we need to equip the models with logical reasoning capabilities (e.g., ability to draw logical conclusions from given statements), which is also a long sought-after goal of AI (Newell & Simon, 1956; McCarthy et al., 1960). One related task is *natural language inference* whose goal is to assign the logical relationships (`contradicted`, `neutral` and `entailment`) to sentence pairs. To push the development of models in logical reasoning, the researchers have focused on more challenging reading comprehension tasks, which often require more complex reasoning as well as longer input. However, most existing reading comprehension datasets are not oriented for the logical reasoning (e.g., SQuAD and HotpotQA), with the exception of LogiQA (Liu et al., 2020), ReClor (Yu et al., 2020) and LR-LSAT (Wang et al., 2021).

Above datasets (LogiQA, ReClor and LR-LSAT) are limited and inadequate to train and evaluate logical reasoning capability. The reason is that all of the three datasets involve diverse types of logical reasoning, such as drawing an alternate conclusion to the argument, finding necessary/sufficient assumptions, whether statements strengthen/weaken the argument or explain/resolve the situation. Mixing multiple types of logical reasoning may pose the following challenges. a). From the perspective of human cognition, different types of logical reasoning correspond to different problem-solving ideas. But in practice we usually train a model on the whole dataset with the same idea, which makes it more limited for models to learn different logical reasoning capability. b). From the perspective of machine learning, if there are many reasoning types mixed in a dataset, then there will be less data for each reasoning type, which is inadequate to train and evaluate logical reasoning capability (demonstrated in our experiments). Furthermore, when the model does not work, it is difficult to determine which reasoning type is the bottleneck (no reasoning type annotation in the dataset), which may hinder the design of better models.

To tackle the challenges, we focus on a more fine-grained type of logical reasoning, named naïve logical reasoning (Section 2), which is to infer the logical conclusion from statements that describe triples (subject, predicate, object) and the relationships among them. A typical example of naïve

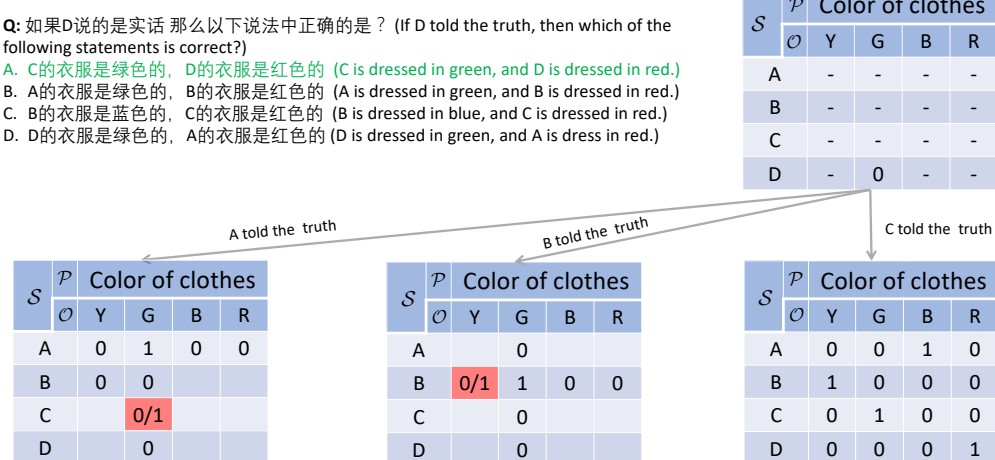

Figure 1: An example of NAIL requiring naïve logical reasoning. Highlighted option A is the correct answer. Tables depict the reasoning process from the human perspective. - indicates uncertain state. 0 and 1 indicate false and true respectively. Red grid with "0/1" means there is a conflict.

logical reasoning is shown in Figure 1. To answer this query, we need to iteratively derive conclusions according to the conditions, and stop the searching branch if a conflict occurs. Specifically, in the initial table, we only know that "D is not green" from "D told the truth". Assume that "A told the truth", and infer that "A is green, C is non-green". Next derive "C is green" from "C told the lie". Then C is derived as both green and non-green, thus causing a conflict. Similar processes for assuming "B told the truth" and "C told the truth". It takes extensive training and practice for human brains to cope with such complex logical reasoning.

Inspired by the datasets extracted from standardized examinations (Lai et al., 2017; Clark et al., 2018; Liu et al., 2020), we build a new large-scale benchmark, NAIL, by selecting naïve logical reasoning examples from standardized exams such as the National Civil Servants Examination of China and Law School Admission Test. However, such examples are limited in their number, as it takes efforts for human experts to design these questions. To collect more data, we propose to imitate examples of standardized exams rather than designing them from scratch. Unlike simple data augmentation (e.g., substitution, paraphrasing), human imitation aims at getting more diverse examples while maintaining the underlying logic (Figure 1). NAIL is available in both Chinese and English, containing a total of $10,296 * 2$ instances.

Empirical results show that state-of-the-art neural models struggle on NAIL with very poor accuracy (the best result is 30.1% for NAIL and 36.2% for Chinese NAIL), while human experts can perform 100% accuracy. Further results indicate that human imitations can significantly help models learn naïve logical ability from natural text.

## 2 NAÏVE LOGICAL REASONING

The naïve logical reasoning is a more fine-grained type of logical reasoning. Formally, we give the definition as follows.

**Definition 1.** *The subject is a described resource, usually an entity, such as person or location. The predicate indicates an attribute of the subject or indicates some kind of relationship between the subject and the object. When denoting an attribute, the object is the attribute value, usually a literal value, e.g., (Jacket, Color, Red), otherwise the object is an entity, e.g., (Beijing, Capital, China).*

**Definition 2.** *Assuming the set of subjects $\mathcal{S}$, the set of predicates $\mathcal{P}$, the set of objects $\mathcal{O}$ and several statements, each statement describes a triple (subject, predicate, object) or some logical relationship*

*between triples. The **naive logical reasoning** is the process of reasoning from these statements to reach a logical conclusion.* [1]

In this work, we explore naïve logical reasoning in the form of reading comprehension. A typical example and detailed reasoning process is shown in Figure 1. Similar to the format of multiple-choice reading comprehension, it contains a context, a query and four options with only one correct answer. To solve the problem, the model needs to understand the logical connections between the subjects, predicates and objects, and then derive a valid option.

## 3    NAIL: DATA COLLECTION AND ANALYSIS

NAIL is a carefully designed benchmark for naïve logical reasoning similar to the format of multiple-choice reading comprehension. Inspired by the datasets extracted from standardized examinations (Lai et al., 2017; Clark et al., 2018; Liu et al., 2020), we first collect a small amount of examples from examinations, denoted as NAIL-E (Section 3.1). Then we propose to imitate examples of NAIL-E to collect more data, denoted as NAIL-I (Section 3.2). Finally we provide a detailed analysis of the proposed NAIL (Section 3.3).

### 3.1    COLLECTION FROM EXAMINATION

We searched for such examples widely from two different types of public examinations: Chinese National Civil Servants Examination (CNCSE) and Law School Admission Test (LSAT). [2] CNCSE is a once-a-year competitive examination in China, and there are overall 120-140 examples per exam per year. But only 1-4 examples belong to the scope of naïve logical reasoning, which is also the most difficult type of problems for candidates within the given 120 minutes. And the LSAT is a standardized test for prospective law school candidates in the United States, Canada, and a growing number of other countries. *Logical Reasoning* is a multiple-choice section of LSAT, containing 24-26 questions under the limitation of 35 minutes, where about 2-4 problems fall into naïve logical reasoning category.

We artificially selected examples that belong to the naïve logical reasoning category from the above two examinations. Finally we obtained 488 examples from the last 25 years of CNCSE and LSAT in the last 30 years, denoted as NAIL-E. These two exams are from countries with different native languages. The former is expressed in Chinese, and the latter is expressed in English. And note that there are slight difference in language style between them. And for diversity and fairness, in later step we get all examples in both Chinese and English through translation.

### 3.2    COLLECTION FROM IMITATION

After collecting a small amount of NAIL-E, we expect to expand the number of the dataset. Designing examples from scratch requires a huge effort from human experts. One simple solution is data augmentation, which artificially scales up data by creating modified data from existing data, such as word/sentence shuffling, word replacement and syntactic variation. However, the data augmented in this way is highly correlated with the original data, and the model easily captures these semantic surface correlations, which makes it limited to train and evaluate logical reasoning capability. To alleviate this limitation, we propose to imitate examples of NAIL-E, to create more examples with a diverse semantic surface while keeping the underlying logic of the original example. Furthermore, in the process of human imitation, we design strict strategies to control the quality of imitation.

**Imitation Example**    See Figure 2, the context of an example from NAIL-E consists of 6 sentences and a query (split by blank lines), each of which can be represented as a logical template (see "backbone template"). The example focus on the description of a scenario: `picking Prince Charming` ($m_1$), which involves five entities: `Li_Na` ($s_1$), `Wang_Wei` ($p_1$), `Wu_Gang` ($p_2$), `Li_Qiang` ($p_3$), `Liu_Dawei` ($p_4$), and three noun/adjective properties: `tall` ($a_1$), `handsome` ($a_2$), `a_PhD` ($a_3$). An accepted imitation needs to keep original underlying logic but have semantically different groundings, i.e., to describe a completely different scenario. Imitation 1 and 2

---

[1]The term *naive* refers to the fact that this logical reasoning process of human in this task is spontaneous, intuitive and unsystematic.

[2]https://www.lsac.org/lsat/taking-lsat/test-format/logical-reasoning

| Backbone Template | Original Example | Imitation 1 ☹ | Imitation 2 ☹ | Imitation 3 ☺ |
|---|---|---|---|---|
| $s_1$心中的 $m_1$ 有如下特征：$a_1$, $a_2$, and $a_3$. ($s_1$'s $m_1$ is supposed to be: $a_1$, $a_2$, and $a_3$.) | 李娜心中的白马王子有如下特征: 高个子、相貌英俊、博士。(Li Na's Prince Charming is supposed to be: tall, handsome, and a PhD.) | 孙怡心中的白马王子有如下特征: 高个子、相貌英俊、博士。(Sun Yi's Prince Charming is supposed to be: tall, handsome, and a PhD.) | 孙怡心中的白马王子有如下特征: 有钱、有房、有车。(Sun Yi's Prince Charming is supposed to be: rich, has a house, and has a car.) | 小丽心中的理想礼物有如下特征: 昂贵，有收藏价值，美观。(Xiao Li's ideal gift is supposed to be: expensive, valuable to collect, and nice-looking.) |
| 她认识 $p_1, p_2, p_3, p_4$ 4（量词）（$p$的类别），其中有一符合$s_1$所要求的全部条件。(She knows 4 *type_of*($p_i$), $p_1, p_2, p_3, p_4$, and only one meets all $s_1$'s requirements.) | 她认识王威、吴刚、李强、刘大伟4位男士,其中有一位符合她所要求的全部条件。(She knows 4 men, Wang Wei, Wu Gang, Li Qiang and Liu Dawei, and only one meets all her requirements.) | 她认识瞿衡、梅震、季凡、张磊4位男士,其中有一位符合她所要求的全部条件。(She knows 4 men, Qu Heng, Mei Zhen, Ji Fan, and Zhang Lei, and only one meets all her requirements.) | 她认识瞿衡、梅震、季凡、张磊4位男士,其中有一位符合她所要求的全部条件。(She knows 4 men, Qu Heng, Mei Zhen, Ji Fan, and Zhang Lei, and only one meets all her requirements.) | 她知道手链、手表、手镯、戒指这4个礼物,其中有一个符合她所要求的全部条件。(She knows 4 gifts, bracelets, watches, bangles, and rings, and only one meets all her requirements.) |
| 4（量词）（$p$的类别）中,有3（量词）是 $a_1$ ,有2（量词）是 $a_3$, 有1（量词）是$a_2$。(Among the 4 *type_of*($p_i$), 3 are $a_1$, 2 are $a_3$, and 1 is $a_2$.) | (1)4位男士中,有3个高个子,2名博士,1人长相英俊。(Among the 4 men, 3 are tall, 2 are PhDs and 1 is handsome.) | (1) 4位男士中,有3个高个子,2名博士,1人长相英俊。(Among the 4 men, 3 are tall, 2 are PhDs and 1 is handsome.) | (1) 4位男士中,有3个有钱,2人有车,1人有房。(Among the 4 men, 3 are rich, 2 have a car and 1 has a house.) | (1) 4个礼物中,有3个很昂贵,2个很美观,1个具有收藏价值。(Among the 4 gifts, 3 are expensive, 2 are nice-looking and 1 is valuable to collect.) |
| $p_1$和 $p_2$都是 $a_3$。( $p_1$and $p_2$are both $a_3$.) | (2)王威和吴刚都是博士。(Wang Wei and Wu Gang are both PhDs.) | (2) 瞿衡和梅震都是博士。(Qu Heng and Mei Zhen are both PhDs.) | (2) 瞿衡和梅震都有车。(Qu Heng and Mei Zhen both have a car.) | (2) 手链和手表都很美。(Bracelets and watches are both nice-looking.) |
| $p_4$和 $p_3$ 关于$a_1$的属性相同。($p_4$and $p_3$are of the same *attribute_of* $a_1$.) | (3)刘大伟和李强身高相同。(Liu Dawei and Li Qiang are of the same height.) | (3) 张磊和季凡身高相同。(Zhang Lei and Ji Fan are of the same height.) | (3) 张磊和季凡有同等程度的钱。(Zhang Lei and Ji Fan are of the same wealth.) | (3) 戒指和手镯价格相同。(Rings and bangles are of the same price.) |
| $p_3$和 $p_1$并非都是 $a_1$ 。(Either $p_3$ or $p_1$ is $a_1$.) | (4)李强和王威并非都是高个子。(Either Li Qiang or Wang Wei is tall.) | (4) 季凡和瞿衡并非都是高个子。(Either Ji Fan or Qu Heng is tall.) | (4) 季凡和瞿衡并非都很富有。(Either Ji Fan or Qu Heng is rich.) | (4) 手镯和手链并非都是昂贵的。(Either Bangles or bracelets is expensive.) |
| 请问谁符合$s_1$ 要求的全部条件？(Who meets all the requirements of $s_1$?) | 请问谁符合李娜要求的全部条件？(Who meets all the requirements of Li Na?) | 请问谁符合孙怡要求的全部条件？(Who meets all the requirements of Sun Yi?) | 请问谁符合孙怡要求的全部条件？(Who meets all the requirements of Sun Yi?) | 请问哪个符合小丽要求的全部条件？(Which meets all the requirements of Xiao Li?) |
| A. $p_4$
B. $p_3$
C. $p_2$
D. $p_1$ | A.刘大伟(Liu Dawei)
B.李强(Li Qiang)
C.吴刚(Wu Gang)
D.王威(Wang Wei) | A.张磊(Zhang Lei)
B.季凡(Ji Fan)
C.梅震(Mei Zhen)
D.瞿衡(Qu Heng) | A.张磊(Zhang Lei)
B.季凡(Ji Fan)
C.梅震(Mei Zhen)
D.瞿衡(Qu Heng) | A.戒指(Rings)
B.手镯(Bangles)
C.手表(Watches)
D.手链(Bracelets) |

Figure 2: An original example from CNCSE and its three imitations, which share the same backbone template. Imitation 1 is subject-level. Imitation 2 is subject-and-object-level. And Imitation 3 is subject-and-predicate-and-object-level.

are unqualified imitations. Since Imitation 1 only conducts *subject*-level imitations (S imitations), i.e., only superficially substitute the five entities ($s_1$:Li_Na→Sun_Yi, $p_1$:Wang_Wei→Qu_Heng, $p_2$:Wu_Gang→Mei_Zhen, $p_3$:Li_Qiang→Ji_Fan, $p_4$:Liu_Dawei→Zhang_Lei). Further, Imitation 2 conducts subject-and-object-level imitations (SO imitations), i.e., not only substituting the entities, but also altering the corresponding objects (property values). ($a_1$:tall→rich, $a_2$:handsome→having_a_house, $a_3$:a_PhD→having_a_car). The difference between Imitation 2 and the original example is much greater than that between Imitation 1 and and the raw problem, however, Imitation 2 is still not an ideal imitation. Furthermore, Imitation 3 changes the scenario $m_1$ into picking ideal gift. Imitation 3 is an expected imitative writing, which conducts subject-and-predicate-and-object-level imitations (SPO imitations). Note that all of the three imitations share the same logic templates. And $p_2$ is the answer for all of the original example and above three imitations.

**Imitation Process** We first select a group of people from a variety of occupations: professional editors, legal practitioners, in-service civil servants and college students (from different majors). Empirically, people in these occupations have strong logical and verbal skills. These people are asked to conduct SPO imitations based on original examples from NAIL-E, and reasonable imitations should meet two requirements: *logic invariance* and *semantic diversity*. We trained these candidate people how to conduct SPO imitations such as Figure 2. We then conducted a trial phase before the official imitation phase, in which process we eliminated some people of poor quality. Finally we employed 82 qualified people[3] to imitatively construct problems based on NAIL-E, and they are paid RMB¥2.8 per imitation[4]. Averagely, it costs a trained person about 3-4 minutes to finish an imitation: starting from coming up a scenario, then replacing the subjects, predicates, and objects of the raw

---

[3]Consisting of 42 native Chinese speakers and 40 native English speakers. Native speakers of a language will be assigned imitation tasks expressed by that language.

[4]A part-time employee can produce 15-20 imitations per hour, where he/she can get RMB¥42-RMB¥56, while the local minimum wage is RMB¥23 per hour.

problem with those scenario-related while keeping invariant logic, and finally smoothing the new sentence with some transition words if necessary. For each original example in NAIL-E, we expect at least 20 imitations (except for extremely difficult cases). Overall we use 813 paid work hours in total to build the NAIL-I.

**Imitation Quality Control** We adopt following strategies to ensure the quality of imitations:

1. As mentioned above, we conducted a trial phase before the official imitation phase. In this phase, we asked them to imitate a small number of problems. Although we do not necessarily need them to write the backbone template, we will check the logic and give feedback to help them understand the task. This process was iterated for three rounds. Only those who passed the trial phase could participate in the official imitation.

2. During the official imitation, we set up an online chat room to communicate with employees and answer their questions timely.

3. To embrace *semantic diversity*, each original example is shown to at least 5 people, that is, one can only conduct 4 imitations based on one original example. People who are assigned to the same example imitate independently without interference from each other, to ensure varied inspiration for imitation.

4. To ensure *logic invariance*, we adopt a double-checking strategy:

   - **Cross Checking**: Everyday, for each employee, we sample 5 imitations from all of his/her daily imitations. And the sampled imitation is assigned to other 2 employees for cross-checking. The imitation will be qualified only if they both approved, and the criterion for approval is that the imitation share the same logics with the original example. If any one of the 5 imitations produced by one employee fail, then all imitations of that employee for that day will be returned to re-check and repair.
   - **Post Checking**: To further ensure that the underlying logic do not deviate during imitation, we introduce another team of experts to solve the imitative examples. The team consists of 20 experienced experts. 10 of them speaks Chinese as native language and have passed CNCSE, while the other 10 speaks English and have passed LSAT.[5] Each imitative example was presented to 3 experts randomly, who are allowed to select one and only choice from "A", "B", "C", "D", otherwise, "UNABLE_TO_ANSWER" if bugs exist in the example, causing no correct choice or multiple correct choices. As long as one of the 3 experts pointed out "UNABLE_TO_ANSWER", then the imitator of this problem should recheck the logic, until each of these 3 experts could give a choice from "A", "B", "C", "D". Note that in the post checking process, we broke up the imitations together and shuffle randomly, otherwise if a person is faced with imitations from same original example, he/she is prone to give a shortcut option with speculation.

**Translation Quality Control** After collecting high-quality mono-collections, we first adopted Google Translation to translate Chinese/English collections into another language, and then employed 10 professional bilingual experts in Chinese and English for manual correction. Bilingual experts were asked to pay attention to logic-invariance and faithfulness during translation. Next, to ensure translation quality, we also adopted the post checking strategy. That is, we asked the 20 human experts mentioned above to solve the translated examples. Each translated example was presented to 3 experts randomly. Since human experts excel in solving naïve logical reasoning problems, (i.e. achieve 100% accuracy on mono-collections), if any expert made a mistake on a translated sample or pointed out "UNABLE_TO_ANSWER", the translated instance is sent back to the bilingual experts for revision. Finally, we asked 50 native speakers to read through all paragraphs of the translation parts in NAIL and mark "0"/"1" for each, where "1" stands for a translated sample is idiomatic, and "0" otherwise. Then for all samples marked with "0" (about 20%), the bilingual experts and native speakers will work together to polish them and conform to the target language norms.

**Human Evaluation** As mentioned above, an imitation is finally regarded as qualified only if the sampled 3 experts could all solve the example. For any original example in NAIL-E, we also ask 3 experts in the team to solve it. Since the gold answers to examples in NAIL-E are provided in public by the examination committee, and corresponding imitations in NAIL-I share the same answer with the original example. Therefore, we calculated the mean accuracy of these three submissions on the overall NAIL, denoted as the performance of **human experts**. To better demonstrate the

---

[5]Experts who are native English speakers will be assigned English problems and experts who are native Chinese speakers will be assigned Chinese problems.

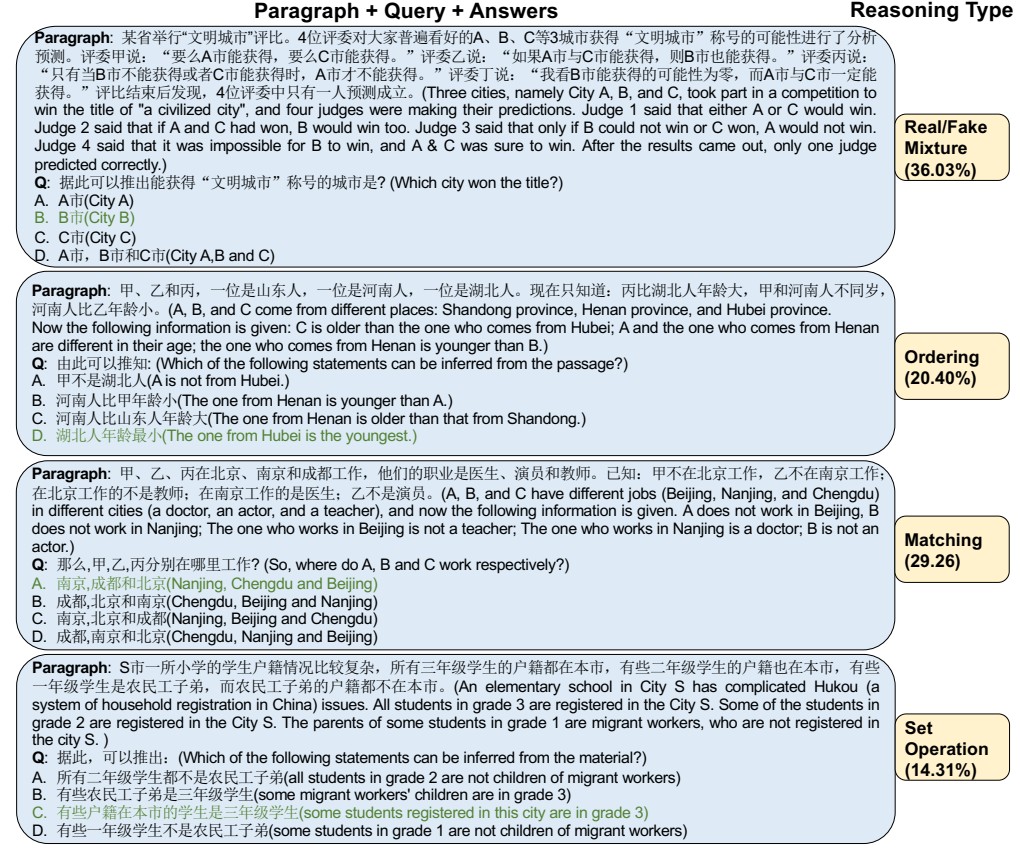

Figure 3: The percentage and representative examples of each naïve logical reasoning type.

difficulty of NAIL, we also selected another team consisting of 20 first-year college students. Same as above, an example is shown to 3 students randomly and they have to give the answer independently. We calculated the overall mean accuracy as the performance of **human baseline**.The evaluation of human baseline is paid separately. Each person can receive RMB¥1.5 for answering each example, which generally cost the person 2-3 minutes.

Since we hire part-time people to write imitatively rather than using existing crowd-sourcing platforms, we build our own website, where the quality and overall progress can be viewed at any time.

## 3.3 DATA ANALYSIS

As mentioned above, NAIL-E and NAIL-I together compose NAIL. NAIL-E, NAIL-I and NAIL are divided into training set, validation set and test set respectively. The overall statistics of NAIL are summarized in Table 1. It is worth noting that an original example in NAIL-E and its corresponding imitations will only be in the same data split. We analyze and manually annotate the fine-grained types of examples in the NAIL-E and group them into 4 categories including real/fake mixture, ordering, matching and set operation, whose percentages and representative examples are shown in Figure 3. Each of these types of examples requires naïve logical reasoning.

| | **Train**(NAIL) | | **Dev**(NAIL) | | **Test**(NAIL) | |
| | NAIL-E | NAIL-I | NAIL-E | NAIL-I | NAIL-E | NAIL-I |
|---|---|---|---|---|---|---|
| # Example | 292 | 5906 | 97 | 1904 | 99 | 1998 |
| # Ave./Max. of paragraph | 68.7 / 155 | 70.2 / 188 | 67.3 / 151 | 70.3 / 212 | 67.0 / 149 | 73.3 / 185 |
| # Ave./Max. of query | 8.1 / 74 | 6.9 / 81 | 9.3 / 71 | 9.3 / 134 | 7.9 / 92 | 7.6 / 71 |
| # Ave./Max. of option | 9.2 / 74 | 9.7 / 130 | 9.1 / 65 | 8.5 / 134 | 8.6 / 76 | 7.7 / 97 |

Table 1: Data split and corresponding statistics.

# 4 EXPERIMENTS

We adopt the *accuracy* as the evaluation metric. Simple **rule-based** methods and strong **neural-based** methods are included as our baseline. Rule-based methods involve *text matching* and *sliding window*. Neural-based methods include BERT (Devlin et al., 2019) and RoBERTa (Liu et al., 2019). Detailed descriptions (e.g., hyper-parameters) of the baseline models and the results on the validation set are listed in the Appendix A. [6]

## 4.1 RESULTS AND ANALYSIS

The performance of all baselines on the NAIL is presented in Table 2. In particular, the human baseline is 71.3 and 76.4 on the test set of NAIL and Chinese NAIL separately, while the human expert is 100.0 since ambiguous examples are not included in the dataset. In comparison, all of the baselines perform much worse than humans, demonstrating that these methods are very limited in naïve logical reasoning. In addition, the results are relatively similar on the English and Chinese datasets. The rule-based approaches achieve an accuracy of 25.8 and 24.2, which is similar to random guess, indicating that word correlation can not be used to help improve performance. Pre-trained mod-

| Model | Input | NAIL | | Chinese NAIL | |
|---|---|---|---|---|---|
| | | Dev | Test | Dev | Test |
| Random | (C, Q, A) | 25.0 | 25.0 | 25.0 | 25.0 |
| Word Matching | (Q, A) | 23.1 | 24.6 | 23.3 | 24.9 |
| | (C, Q, A) | 21.6 | 25.8 | 22.2 | 24.2 |
| Sliding Window | (Q, A) | 23.8 | 24.2 | 24.1 | 24.7 |
| | (C, Q, A) | 21.9 | 22.1 | 21.6 | 22.5 |
| BERT LARGE | (A) | 26.8 | 26.7 | 27.2 | 26.7 |
| | (Q, A) | 27.3 | 27.1 | 27.4 | 26.8 |
| | (C, Q, A) | 29.0 | 29.4 | 29.3 | 27.7 |
| RoBERTa LARGE | (A) | 27.3 | 26.5 | 29.4 | 27.7 |
| | (Q, A) | 27.8 | 27.9 | 32.6 | 30.5 |
| | (C, Q, A) | 34.6 | 30.1 | 37.3 | 36.2 |
| Human baseline | (C, Q, A) | 70.1 | 71.3 | 75.5 | 76.4 |
| Human expert | (C, Q, A) | 100.0 | 100.0 | 100.0 | 100.0 |

Table 2: Main results on NAIL (accuracy%). The column Input means whether to input context (C), query (Q) and answer options (A).

els have relatively good performance with about 4-10 points improvement compared to the rule-based approaches, showing that pre-trained models have a certain degree of commonsense and logical reasoning capabilities (Huang et al., 2019b). However, the best result by RoBERTa LARGE is 36.2 on the testing data of Chinese NAIL, which is still far below the human performance. This indicates that the knowledge in the pre-trained model is quite weak in logical reasoning.

**Different Input Settings** We conduct experiments with different input settings. The setting of questions and answer options (Q, A) does not lead to significant improvements compared to the input setting of only answer options (A). One likely reason is that the queries usually do not provide much information, e.g., *According to this, it can be deduced that?* Further adding context yields a noticeable boost, showing that the context is highly informative.

**Transfer Learning** We conduct a set of transfer learning experiments to understand the degree of overlap in terms of necessary knowledge for solving problems in our dataset and existing datasets.

*What are the results if pre-trained model is first trained on existing reading comprehension datasets, and then fine-tuned on NAIL?* Table 3 shows the results where LogiQA, ReClor and RACE are adopted. [7] Overall, we observe that when LogiQA, ReClor or RACE is regarded as extra training resource for RoBERTa, the performance on NAIL will increase (30.1→34.5/33.4/31.0), since models can learn some degree of general reasoning ability when learning other comprehension tasks. While interestingly, if models are trained only on LogiQA, ReClor, or RACE, and then zero-shot to directly test on NAIL, the performance is poor and close to random guess. And as for RACE, the zero-shot performance on NAIL (22.0) is even far below than that of RoBERTa (23.4). We attribute this to the

---

[6]We also try seq2seq-based models (i.e. T5 (Raffel et al., 2019)) and well-designed models for other related tasks (i.e., DAGN (Huang et al., 2021) for LogiQA and ReClor, HGN (Fang et al., 2020) for HotpotQA (Yang et al., 2018), and QDGAT (Chen et al., 2020) for DROP (Dua et al., 2019)) on our English version of NAIL. Details and results are shown in Appendix B.

[7]For fair comparison, in all cross-benchmarks experiments in this paper, we removed samples in the training set that are duplicates of those in the test set.

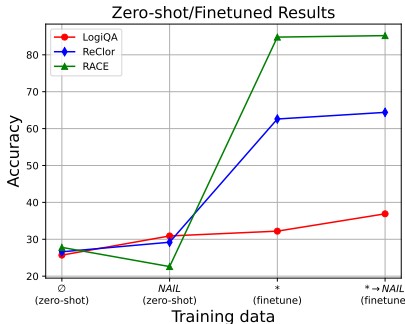

Figure 4: Transfer learning results when evaluating on the test split of LogiQA, ReClor and RACE. ∅ is the zero-shot performance of RoBERTa. * is a placeholder for these three datasets.

| Evaluate on → Train on ↓ | NAIL | NAIL-E | NAIL-I |
|---|---|---|---|
| ∅ | 23.4 | 20.8 | 23.5 |
| LogiQA | 25.4 | 39.6 | 24.7 |
| ReClor | 24.7 | 19.8 | 25.0 |
| RACE | 22.0 | 29.2 | 21.6 |
| NAIL | 30.1 | 37.4 | 29.7 |
| LogiQA→NAIL | 34.5 | 38.5 | 34.3 |
| ReClor→NAIL | 33.6 | 33.3 | 33.4 |
| RACE→NAIL | 31.0 | 32.3 | 31.0 |

Table 3: Transfer learning results when evaluating on the test split of NAIL, NAIL-E, NAIL-I. RACE → NAIL denotes finetuned on RACE first and then finetuned on NAIL.

fact that, naïve logical reasoning is a typical category of reasoning which should be significantly distinguished from RACE. Fine-tuning on RACE makes the parameters fit the RACE data, which can lead to side effects on NAIL. We believe that completely different categories of reasoning deserve to be explored separately since they may have different forms or strategies of reasoning, especially when training data is insufficient.

*What are the results if* NAIL *is used as extra training resource for existing reading comprehension tasks?* Figure 4 shows answers to this question. Generally, we can draw the conclusion that using NAIL as a pre-training step can significantly improve the supervised-learning performance for other tasks, such as LogiQA (35.3→36.9 for test), ReClor (62.6→64.4),[8] and RACE (83.2→85.2). This indicates that NAIL can bring naïve logical reasoning ability to the model, which is a basic reasoning ability and can be reflected into other comprehension tasks. An interesting thing is that for zero-shot evaluating on RACE, NAIL seems to have a side effect in the pretraining process (22.6<27.8). This is because of a similar reason as mentioned above, that NAIL and RACE consist of completely different categories of examples. Results of more datasets are shown in Appendix D.2.

**Fine-grained Types** We further analyze the model performance with respect to different types of naïve logical reasoning (Figure 5). We find that language models perform well on set operation problems, while struggle on matching and ordering. We think that language models can provide good representation of set object, even if models do not really reason derived from the context.

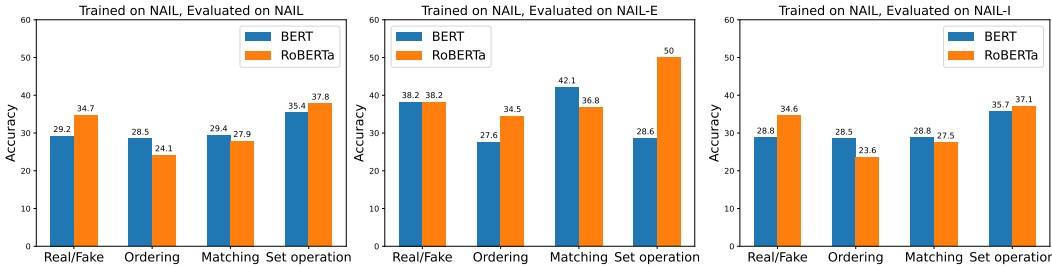

Figure 5: Accuracy against reasoning types evaluated on different testing sets.

**Error Analysis** We further perform detailed analysis of human-baseline errors and model errors, while from Table 2, we can observe that human baseline on NAIL (around 70%) is much lower than that of human expert (100%), and RoBERTa performs much worse (above 30%). We measure the accuracy against several factors on the test set of NAIL: *number of sentences in the backbone template of the context , number of possible worlds,* [9] *and context length.*

See Figure 6(a), as the number of sentences in the backbone template decreases, the accuracy rate of human baseline increases significantly, and when reduced to 2 and below, human baseline does

---

[8]We report the result on the validation set of ReClor, since gold answers for test are not acquired.

[9]The number of possible worlds is denoted as, *the number of subjects × the number of predicates × the number of objects*.

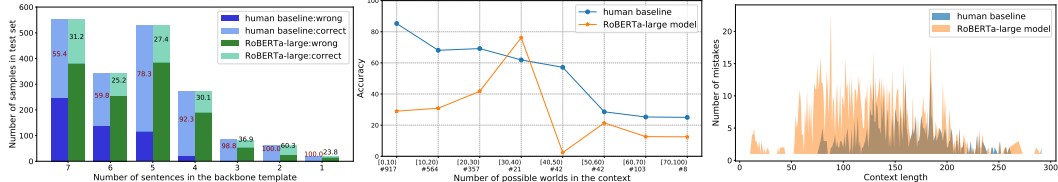

Figure 6: (a). X-axis: the number of sentences in the backbone template. The height of bars: Number of samples in the test set grouped by X-axis. The number in bars: The accuracy against X-axis. (b). Accuracy of human baseline and RoBERTa against the number of possible worlds in the context.("$[a, b)\#k$" in the x-axis denotes there are $k$ samples in the test set where $a \leq number\_of\_possible\_worlds < b$. (c). Number of mistakes made by human baseline and RoBERTa model against various context length.

not make any mistakes, while performance of RoBERTa does not show any trends, even when the number equals to 1, the model still makes mistakes in a large percentage of cases. See Figure 6(b), as the number of possible worlds in the context increases, the accuracy of human baseline tends to decrease significantly. This is because the more possible worlds that need to be considered, the more judgments humans need to perform, and humans tend to overlook certain conflicts, which leads to wrong decisions for problems. However there is no significant correlation between the performance of the model and number of possible worlds. This indicates that the model does not really filter and judge the possible worlds. See Figure 6(c), the model makes mistakes at a variety of context lengths, while humans perform perfectly for problems with context length $\leq 73$.

## 5 RELATED WORK

There are abundant datasets in reading comprehension, which could facilitate the development of the field. MCTest (Richardson et al., 2013) is a multiple-choice reading comprehension dataset that contains 500 fictional stories and 2k questions. Rajpurkar et al. (2016) proposes the first large-scale reading comprehension dataset SQuAD (100k+ questions), where the answer to each question is a span of text from the passage. Recently more datasets requiring more complicated reasoning types are introduced, such as multi-document (Joshi et al., 2017; Dunn et al., 2017), multi-hop reasoning (Yang et al., 2018; Welbl et al., 2018; Talmor & Berant, 2018), numerical discrete reasoning (Dua et al., 2019) and commonsense reasoning (Mihaylov et al., 2018; Zhang et al., 2018; Huang et al., 2019a). However, these datasets cannot test the logical reasoning ability of the models. To fill this gap, LogiQA (Liu et al., 2020), ReClor (Yu et al., 2020) and LR-LSAT (Wang et al., 2021) was proposed.

LogiQA is collected from National Civil Servants Examination. ReClor and LR-LSAT are extracted from Law School Admission Test. These datasets require logical reasoning to answer the questions. However, from human experience, there are different forms of reasoning strategies in answering different logical questions. We believe that completely different categories of logical reasoning deserve to be explored separately (Rudinger et al., 2020). Different from these datasets, we inductively define a typical class of logical reasoning, named naïve logical reasoning, and then create a new benchmark targeting the task, named NAIL. Compared with LogiQA, Reclor and LR-LSAT, NAIL focus on the more fine-grained logical reasoning type (naïve logical reasoning).

There have been many datasets extracted from human examinations, such as LogiQA and ReClor mentioned above. Besides, RACE dataset (Lai et al., 2017) is collected from English exams for middle and high school Chinese students. ARC dataset (Clark et al., 2018) consists of 7,787 science exam questions drawn from a variety of sources.DREAM (Sun et al., 2019) is dialogue-based multiple-choice reading comprehension dataset collected from English as a Foreign Language examinations which contains 10,197 questions for 6,444 dialogues.

## 6 CONCLUSION

We introduce a more fine-grained logical reasoning, naïve logical reasoning, and We propose a new large-scale benchmark, NAIL, aiming to help models learn and evaluate naïve logical reasoning capability. NAIL is sourced from standardized exams and human imitation. Preliminary results show that there is still a long way to go to equip deep models with true logical reasoning capability.

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

# A   IMPLEMENTATION DETAILS OF BASELINES

**Rule-based Methods**   Following LogiQA  (Liu et al., 2020), we adopt two simple rule-based methods based on text matching. Specifically, *word matching* is to measure the degree of unigram overlap between the candidate answer and the given paragraph-query pair; *sliding window* takes into account the n-gram when calculating the matching score.

**Neural-based Methods**   Pre-trained neural models, such as BERT  (Devlin et al., 2019), RoBERTa (Liu et al., 2019), have achieved impressive performance on reading comprehension. The input to the pre-trained model is the concatenation of the paragraph, the query and the candidate answer, separated by [SEP] tokens, denoted as: [CLS], paragraph, [SEP], query, [SEP], answer. After the encoding of the pre-trained model, the representation of [CLS] token followed by an multiple layer perceptron (MLP) is used for scoring.

We re-implement the rule-based methods following LogiQA  (Liu et al., 2020). For pre-trained models, we modify the code of Transformers of HuggingFace  (Wolf et al., 2020) to implement them on NAIL. We take the off-the-shelf model BERT and RoBERTa for NAIL, and Chinese BERT and Chinese RoBERTa  (Cui et al., 2019) for Chinese NAIL. All hyper-parameters are selected by the model performance on the development sets.

# B   MORE STRONG BASELINES

We also try seq2seq-based pre-trained models (i.e. T5 (Raffel et al., 2019)) and well-designed models for other related tasks (i.e. DAGN (Huang et al., 2021) for LogiQA and Reclor, HGN (Fang et al., 2020) for HotpotQA (Yang et al., 2018), and QDGAT (Chen et al., 2020) for DROP (Dua et al., 2019)) on our English version of NAIL.

A problem is regarded as four samples to T5 during training. The input of a sample is: context+query+one of the four choices. And we let the model generate the label for the sample: if the choice is the correct answer, then generate "True", otherwise generate "False". And during inference, for a problem, we choose the option with the highest probability of generating "True" among the four options as the prediction. We set $input\_max\_len = 512$ and finetuned the off-the-shelf T5-base model on the training set of NAIL, results are shown in Table 4.

LogiQA and ReClor are two logical benchmarks introduced in the text of the paper to test models' various reasoning abilities. HotpotQA is a multiple-choice QA dataset featuring natural, multi-hop questions, with strong supervision for supporting facts to test models' multi-hop factual reasoning ability. DROP is a 96k-question benchmark to test models' numerical discrete reasoning ability over paragraphs (such as addition, counting, or sorting).

We choose three state-of-the-art and open-sourced models designed for the above benchmarks: DAGN for LogiQA and Reclor, HGN for HotpotQA, and QDGAT for DROP respectively.We reproduced these three models with the code released by authors[10] and NAIL as training data. Results on NAIL are shown as Table 4.

| Model | Encoder | NAIL | |
|---|---|---|---|
| | | **Dev** | **Test** |
| RoBERTa $_{LARGE}$ | RoBERTa $_{LARGE}$ | 34.6 | 30.1 |
| RoBERTa $_{BASE}$ | RoBERTa $_{BASE}$ | 27.6 | 26.4 |
| T5 $_{BASE}$ | T5 $_{BASE}$ | 27.1 | 26.3 |
| **DAGN** | RoBERTa $_{LARGE}$ | **36.0** | **32.3** |
| HGN | RoBERTa $_{LARGE}$ | 30.1 | 28.7 |
| QDGAT | RoBERTa $_{LARGE}$ | 28.3 | 28.7 |

Table 4: Results on NAIL (accuracy%).

---

[10]DAGN:  https://github.com/Eleanor-H/DAGN,   QDGAT:  https://github.com/emnlp2020qdgat/QDGAT, HGN: https://github.com/yuwfan/HGN

We observe that DAGN achieves the best results on NAIL, since ReClor and LogiQA are much closer to NAIL, the well-designed discourse-aware graph network DAGN is more applicable to NAIL. However there is still very much space for models' improvement compared to humans. HGN is a hierarchical graph network for multi-hop question answering, which is designed to aggregate heterogeneous information, and QDGAT is a question directed graph attention network, which is dedicated to identifying numbers and the computation between them, which are both of minimal help to NAIL. T5 is a very powerful generative model, but it seems that T5 lacks the naïve reasoning capability as well. Due to computational resource limitation, we did not try a T5 model with larger number of parameters.

## C  THE EFFECT OF NAIL, NAIL-E AND NAIL-I

To investigate the effect of NAIL, NAIL-E and NAIL-I, we train a RoBERTa $_{LARGE}$ model on NAIL, NAIL-E and NAIL-I separately and evaluate on the corresponding set of all tests (Table 5). When trained on NAIL-I and evaluated on NAIL-E, the model achieves the best accuracy 38.5%, showing that human imitations can significantly help models learn logic from natural text. When trained on NAIL-E and evaluated on NAIL-E, the model achieves the worst accuracy 22.9%. The possible reason is that the amount of data of NAIL-E is too small to train a deep model. Corresponding results on the validation set are shown in Table 6.

| Trained on → 
 Evaluated on ↓ | NAIL | NAIL-E | NAIL-I |
|---|---|---|---|
| NAIL | 30.1 | 26.4 | 33.9 |
| NAIL-E | 37.4 | 22.9 | 38.5 |
| NAIL-I | 29.7 | 26.6 | 33.7 |

Table 5: Performance of RoBERTa on different training sets and test sets.

| Trained on → 
 Evaluated on ↓ | NAIL | NAIL-E | NAIL-I |
|---|---|---|---|
| NAIL | 34.6 | 24.9 | 32.0 |
| NAIL-E | 43.3 | 37.4 | 33.0 |
| NAIL-I | 34.2 | 24.3 | 31.9 |

Table 6: Performance of RoBERTa on different training sets and validation sets.

## D  TRANSFER LEARNING

### D.1  OTHER DATASETS AS EXTRA TRAINING RESOURCE

*What are the results if pre-trained model is first trained on existing reading comprehension datasets, and then fine-tuned on NAIL?* Table 3 shows the results on the test set of NAIL,NAIL-E, NAIL-I where LogiQA, ReClor[11] and RACE are adopted. And here in Table 7 we give the results on the validation splits of NAIL,NAIL-E, NAIL-I.

### D.2  NAIL AS EXTRA TRAINING RESOURCE

*What are the results if NAIL is used as extra training resource for existing reading comprehension tasks?* Using NAIL as extra training resource, we conduct plenty of experiments on existing benchmarks. Besides LogiQA, ReClor and RACE (see in Figure 4), more results are shown in

---

[11]For fair comparison, in all cross-benchmarks experiments in this paper, we removed samples in the training set that are duplicates of those in the test set. For example, when using ReClor or LogiQA as extra training resource when testing on NAIL, we remove problems in the training set of ReClor or LogiQA if they appear in the test set of NAIL.

| Evaluate on → Train on ↓ | NAIL | NAIL-E | NAIL-I |
|---|---|---|---|
| ∅ | 23.6 | 24.2 | 23.6 |
| LogiQA | 33.0 | 45.1 | 32.4 |
| ReClor | 28.4 | 25.3 | 28.5 |
| RACE | 26.4 | 27.5 | 26.4 |
| NAIL | 34.6 | 43.3 | 34.2 |
| LogiQA→ NAIL | 37.9 | 47.3 | 37.5 |
| ReClor→ NAIL | 35.2 | 39.6 | 35.9 |
| RACE→NAIL | 35.4 | 36.3 | 35.4 |

Table 7: Transfer learning results when evaluating on the validation split of NAIL, NAIL-E, NAIL-I (accuracy%).

Table 8, 9. We adopted MathQA (Amini et al., 2019), and HotpotQA (Yang et al., 2018). For MathQA, we include a prediction head for multiple choice based on the RoBERTa-large model, and for HotpotQA,[12] we include a prediction head for question answering. For MathQA, we concat the *problem* and *annotated_formula* as input.

| Evaluate on → Train on ↓ | MathQA Test |
|---|---|
| MathQA | 39.8 |
| NAIL → MathQA | 41.2 |

| Evaluate on → Train on ↓ | HotpotQA Ans F1 |
|---|---|
| HotpotQA | 69.8 |
| NAIL → HotpotQA | 72.6 |

Table 8: Evaluating on MathQA after RoBERTa-large pretrained on MathQA/NAIL training set.

Table 9: Evaluating on HotpotQA after RoBERTa-large pretrained on HotpotQA/NAIL training set.

Experimental results show that, if we use NAIL as extra training resource, the supervised-learning results on all of these six datasets: LogiQA, ReClor, RACE, MathQA, HotpotQA will improve. This indicates that equipping models with naïve logical reasoning ability can help solve math word problems (MathQA), improve multi-hop factual reasoning skills (HotpotQA), solve various logical reasoning problems (LogiQA and ReClor), and enhance general understanding and reading comprehension capability (RACE). In the future, we will test on more known datasets, to verify that naïve logical reasoning is a basic capability and is helpful for other tasks.

# E  ANALYSIS OF FINE-GRAINED TYPES

In Section 4, we analyze the model performance with respect to different types of naïve logical reasoning (Figure 5). Figure 8 shows the accuracy against fine-grained reasoning types on the test set of NAIL, NAIL-E, NAIL-I when training on NAIL, NAIL-E, NAIL-I respectively. And Figure 7 shows the corresponding results on the validation set.

From above figures we can find that language models perform well on set operation problems, while struggle on matching and ordering. We think that language models can provide good representation of set object, even if models do not really reason derived from the context.

# F  DISCUSSIONS ABOUT FUTURE DIRECTIONS

We have some ideas for future directions for models to solve NAIL.

The first point is to identify deterministic information or information with low uncertainty. Generally, we need to detect the atomic information that can be used directly without reasoning, which is the key to solving the problem. Different conditions should not be considered with equal priority,

---

[12]Due to computational resource limitations, we only trained 3 epochs for experiments involved HotpotQA.

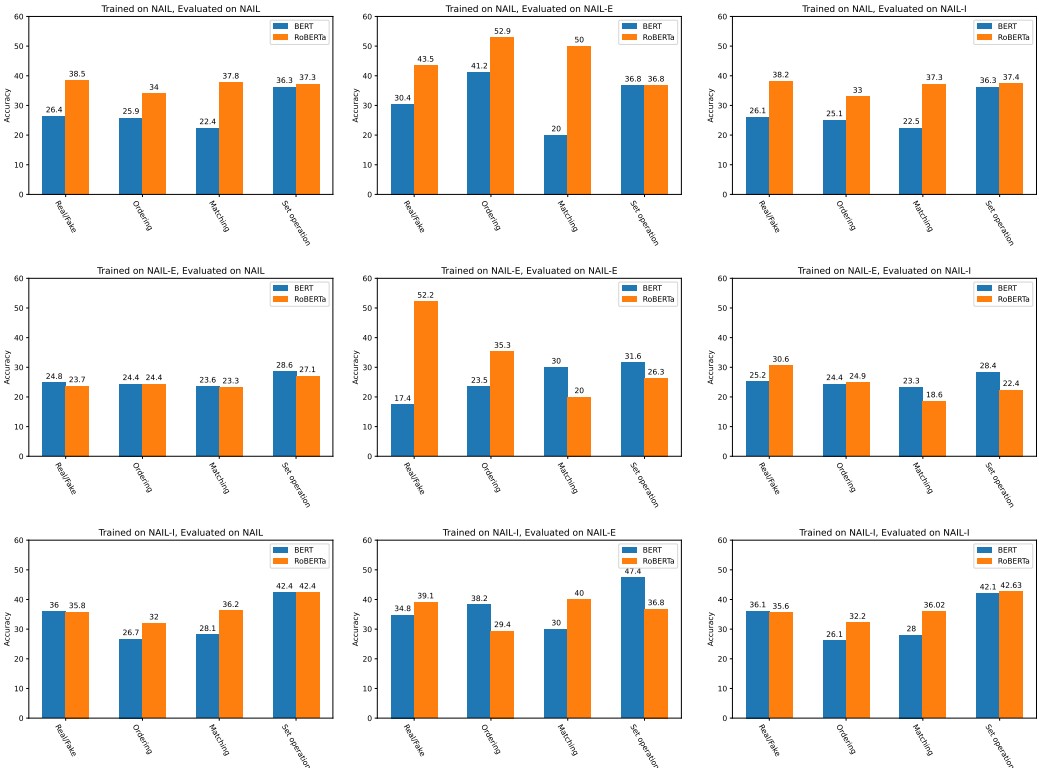

Figure 7: Accuracy against the reasoning types based on different training set and development set.

i.e., the more certain a condition we start from, the fewer possible worlds that the condition yields. Specifically, for real/fake mixture problems, we also need to promptly identify two atomic statements that yield a contradictory, that is, the two statements cannot be both true (there must be one false) or both false (there must be one true), which is often the breakthrough in solving real/fake mixture problems.

The second is to detect informative subjects/objects. Generally, the more frequently a subject/object is mentioned in context, the more relevant information it carries. Tables and bi(/tri)partite diagrams can help to clarify the correspondence between given subjects and objects. For example, in the case of Figure 1, tables play an effective role in solving the problems. And as for the third case in Figure 3, which is categorized as a matching problem, a tri-partite diagram could help, that is, three disjoint and independent sets U, V and W represent {A,B,C}, {Beijing, Nanjing, Chengdu}, {a doctor, an actor, and a teacher} respectively, an edge connecting a vertex in one set to one in another set denotes the "is_a" predicate.

The third direction is allowing models to better utilize elimination. On the one hand, models can perform forward elimination, i.e., according to the context, each time we draw a definite conclusion, we can retain options that are logically consistent and exclude those that do not fit the derived conclusion. On the other hand, models can also perform backward elimination. For example, when sometimes it is not easy to draw exact inferences directly from the context, then we can substitute the options into the context. If substituting an option creates a contradiction within the context, then the option should be excluded. Forward and backward elimination facilitate the model in arriving at the correct answer.

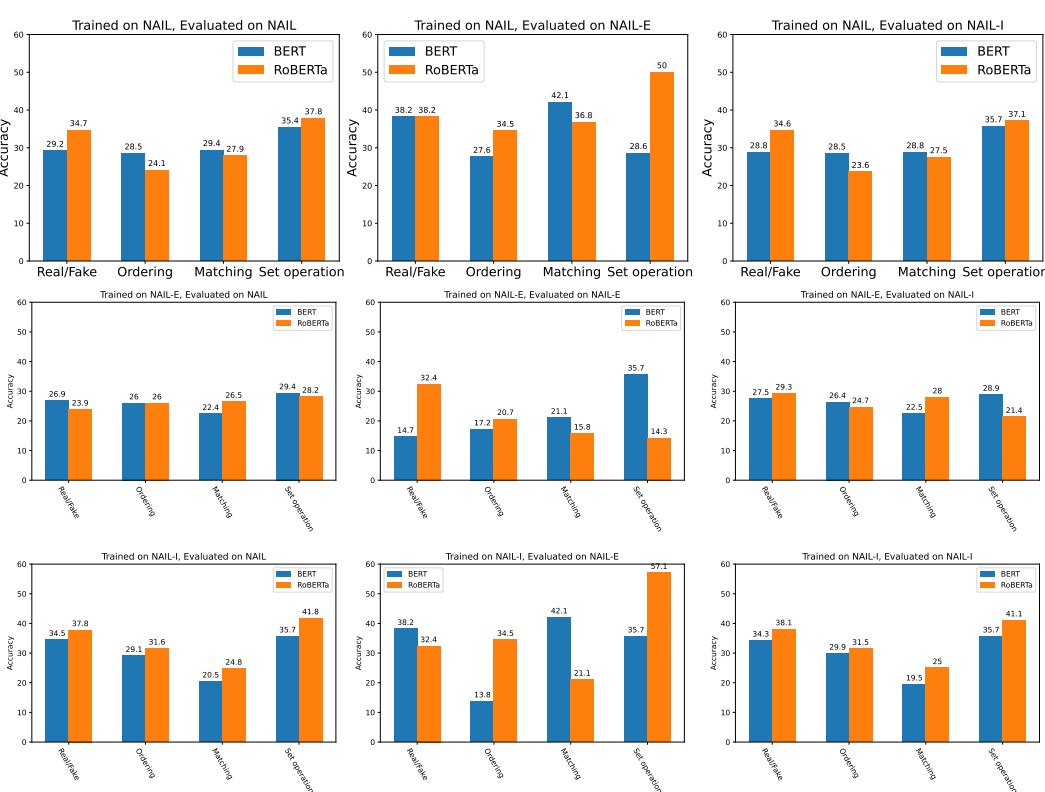

Figure 8: Accuracy against the reasoning types based on different training set and testing set.

