# OpenReview forum: "NAIL: A Challenging Benchmark for Na\"ive Logical Reasoning"
_ICLR.cc/2022/Conference — ICLR 2022 Submitted_

### Official Review · Reviewer_QPXV · 2021-10-31

**Correctness:** 3
**Technical Novelty And Significance:** 3
**Empirical Novelty And Significance:** 2
**Recommendation:** 5
**Confidence:** 3

**Main Review:**

Strengths:

- The proposed dataset is carefully designed and sourced where most of the questions are collected by imitating actual examples in exams. All examples are validated by human experts.
- The collected questions are difficult for BERT-large and RoBERTa-large which show lower accuracies than the human baseline of college students.

Weaknesses:

- The authors test simple rule-based models and vanilla BERT-large/RoBERTa-large models, failing to imply a future direction for developing systems that perform naive logical reasoning. Ideally, the authors could (modify and) test graph neural network-based systems that are proposed for datasets that require symbolic reasoning (e.g., HotpotQA and DROP).
- It is a bit obvious that systems pre-trained on ReClor, LogiQA, and RACE do not perform well on NAIL because, as the author mention in Section 4.2, the nature of questions are completely different. Rather, I would like to see it used as an extra training resource for existing multiple-choice datasets including these three datasets. I agree with the authors that the naive logical reasoning needs to be explored as an independent research topic, but I think the authors need to prove that the proposed dataset is useful in some sense in relation to other NLU tasks, datasets, and systems.
- Likewise, the main claim "human imitations can significantly help models learn logic from natural text" is supported only by the experiments on the actual examples contained in NAIL (NAIL-E). The authors prove that the models trained on the imitation examples can perform specific types of reasoning contained in the actual examples, but do not assess their ability of more general logical reasoning such as in ReClor and LogiQA.
- I'm not sure if the authors can use "naive" in the sense that is referred to in the context of Naive Set Theory by Halmos (1970). I guess the notion "naive" in Naive Set Theory means that the theory is not described on the top of axioms. Does this imply "spontaneous, intuitive, and unsystematic"?
- There are grammatical errors or typos in the paper. It should be proofread.

Questions and feedback:

- It is unclear how the authors translate Chinese and English examples. In particular, if it is done after the expert validation, translation errors might affect the human baseline accuracy. Are there any quality controls over the translation?
- Human baseline (around 70%) looks a bit lower than that of experts (100%). Is there an error analysis about them? Is that by careless mistakes or by being truly difficult for college students?
- The second paragraph of Section 5 sounds interesting but has only one sentence. Could the authors flesh it out?
- In the description of the post-checking process, what does "each of experts has been selected through CNCSE" mean?
- Please replace the alias tokens in the Open Review abstract with actual terms.

**Summary Of The Paper:**

This paper proposes a new dataset that tests the capability of logical reasoning through multiple-choice reading comprehension. Questions are collected from human exams (Chinese National Civil Servants Examination in Chinese and Law School Admission Test in English) and are augmented by imitating the actual exam examples. The main focus of the questions is understanding of the relationship between subject-predicate-object triples, involving four reasoning types: factuality, ordering, matching, and set operation. In the imitation process, the authors hire people from different backgrounds who are asked to write examples that are logically invariant but semantically different from a given actual example (82 people, 813 paid work hours with fair payment,  and at least 20 imitations per actual example). All collected examples are manually validated by experts and answered by college students for measuring human baseline. Experiments with rule-based (word matching and sliding window) and neural network-based (BERT and RoBERTa) models show that the machine performance (30.10 and 36.15 for the English and Chinese sets respectively) is largely lower than the human baseline (71.31 and 76.38), which indicates that these systems struggle to solve the naive logical reasoning questions.

**Summary Of The Review:**

The proposed dataset is carefully designed and may be useful for pushing the boundary of assessing the logical reasoning capability of systems through the reading comprehension task, but experiments are insufficient for empirically showing the usefulness of the dataset.

---

> ### Author Response · Authors · 2021-11-23
> **Response to Reviewer QPXV**
>
> Thanks for your review comments. We briefly respond to your questions as follows.
>
> >**Modify and test GNN-based systems that are proposed for datasets that require symbolic reasoning (e.g., HotpotQA and DROP)**
>
> Thanks for your valuable comments.
> We choose three state-of-the-art and open-sourced models designed for three related benchmarks: DAGN for LogiQA and Reclor, HGN for HotpotQA, and QDGAT for DROP respectively.
> We reproduced these three models with the code released by authors and NAIL as training data:
>
> | MODEL | ENCODER|DEV| TEST|
> |----|----|----|----|
> | RoBERTa-large | RoBERTa-large | 34.6 | 30.1 |
> | DAGN | RoBERTa-large | 36.0 | 32.3 |
> | HGN | RoBERTa-large | 30.1 | 28.7 |
> | QDGAT | RoBERTa-large | 28.3 | 28.7 |
>
> We observe that DAGN achieves the best results on NAIL, since ReClor and LogiQA are much closer to NAIL, the well-designed discourse-aware graph network DAGN is more applicable to NAIL.
> However, there is still very much space for models' improvement compared to humans.
> For more details, please refer to Appendix B in the revised paper.
>
>
>
> >**It is used as an extra training resource for existing multiple-choice datasets, and assess imitation's ability of more general logical reasoning such as in ReClor and LogiQA**
>
> Thanks for your kind advice.
> We conduct such experiments supplementally. We use NAIL as extra training resource for RoBERTa for learning existing reading comprehension tasks, where LogiQA, ReClor, RACE, HotpotQA, MathQA, AQuA are adopted (experiments on CLUTRR, AQuA are ongoing, and the results will be updated in time), and supervised-learning results are listed as follows.
>
> |Number| Training Settings |  Evaluation Settings |Results|
> |----|----|----|----|
> | 1 | LogiQA  |  LogiQA  | 35.3 |
> | 2 | NAIL$\rightarrow$LogiQA | LogiQA  | 36.9 (+1.6)|
> | 3 | ReClor |ReClor  | 62.6 |
> | 4 | NAIL$\rightarrow$ReClor | ReClor  | 64.4 (+1.8)|
> | 5 | RACE  |RACE | 83.2 |
> | 6 | NAIL$\rightarrow$RACE |RACE | 85.2 (+2.0)|
> | 7 | MathQA | MathQA    | 39.8  |
> | 8 | NAIL $\rightarrow$ MathQA |  MathQA | 41.2 (+1.4)|
> | 9 | HotpotQA|HotpotQA    | 69.8  |
> | 10 | NAIL $\rightarrow$ HotpotQA| HotpotQA | 72.6 (+2.8)|
>
> Generally, we can draw the conclusion that using NAIL as a pre-training step can significantly improve supervised-learning performance for other tasks, with boosts ranging from 1.4 points to 2.8 points. This indicates that NAIL can bring naive logical reasoning ability to the model, which is a basic reasoning ability and can be reflected into other comprehension tasks, such as solving math word problems (MathQA), multi-hop factual reasoning (HotpotQA), various logical reasoning (LogiQA and ReClor), and general understanding (RACE).
> More details can be found in "Transfer Learning" in Section 4 and Appendix D.2.
>
> We also conduct zero-shot experiments, that is, using NAIL as training resource, and test directly on the mentioned datasets (LogiQA, ReClor, and RACE).
> For more details, please refer to Figure 4 in "Transfer Learning" in Section 4 and Appendix D.2.
> For more error analysis,  please refer to  "Error Analysis" in Section 4 in the revised paper.
>
>
> >**How the authors translate Chinese and English examples**
>
> Thanks for the nice comments.
> Sorry for misunderstanding how to guarantee the quality of translation, we omitted this due to space limitations.
> In fact, after collecting high-quality mono-collections,
> + We first adopted Google Translation to translate Chinese/English collections into another language.
> + Then employed 10 professional bilingual experts in Chinese and English for manual correction. Bilingual experts were asked to pay attention to logic-invariance and faithfulness during translation.
> + Next, to ensure translation quality, we also adopted the post checking strategy. That is, we asked the 20 human experts mentioned above to solve the translated examples. Each translated example was presented to 3 experts randomly. Since human experts excel in solving naive logical reasoning problems, (i.e. achieve 100% accuracy on mono-collections), if any expert made a mistake on a translated sample or pointed out  "UNABLE_TO_ANSWER'', the translated instance is sent back to the bilingual experts for revision.
> + After that, we asked 50 native speakers to read through all paragraphs of the translation parts in NAIL and mark 0/1 for each, where 1 stands for a translated sample is idiomatic, and 0 otherwise. Then for all samples marked with 0 (about 20%), the bilingual experts and native speakers will work together to polish them and conform to the target language norms.
>
> We added a subsection to the revision of the paper detailing the translation process (see "Translation Quality Control" of Section 3.2).

---

> > ### Author Response · Authors · 2021-11-23
> > **Continue Response to Reviewer QPXV**
> >
> > >**Human baseline (around 70%) looks a bit lower than that of experts (100%)**
> >
> > We make detailed analysis about human-baseline errors in the revised paper.
> > We measure the accuracy against several factors on the test set of NAIL: number of sentences in the backbone template of the context , number of possible worlds, and context length.
> > There are several findings.
> > + As the number of sentences in the backbone template decreases, the accuracy rate of human baseline increases significantly, and when reduced to 2 and below, human baseline does not make any mistakes.
> > + As the number of possible worlds in the context increases, the accuracy of human baseline tends to decrease significantly. This is because the more possible worlds that need to be considered, the more judgments humans need to perform, and humans tend to overlook certain conflicts, which leads to wrong decisions for problems.
> > + Humans perform perfectly for problems with context length $\leq$ 73, and humans are more likely to make mistakes when the context length gets longer.
> >
> > For more details, please check the "Error Analysis" paragraph in Section4 of the revised paper.
> >
> > >**The second paragraph of Section 5 sounds interesting but has only one sentence. Could the authors flesh it out?**
> >
> > Thank you for your interest! We did not describe it in detail due to limited space.
> > In fact, the second paragraph means to detect informative subjects/objects.
> > Generally, the more frequently a subject/object is mentioned in context, the more relevant information it carries.
> > Tables and bi(/tri)partite diagrams can help to clarify the correspondence between given subjects and objects. For example, in the case of Figure 1, tables play an effective role in solving the problems.
> > And as for the third case in Figure 3, which is categorized as a matching problem, a tri-partite diagram could help, that is, three disjoint and independent sets U, V and W represent {A,B,C}, {Beijing, Nanjing, Chengdu}, {a doctor, an actor, and a teacher} respectively, an edge connecting a vertex in one set to one in another set denotes the "is_a" predicate.
> > More details can be found in Appendix F in the revised paper.
> >
> > >**What does "each of experts has been selected through CNCSE" mean?**
> >
> > Sorry for the misunderstanding caused by our writing.
> > We will restate it as "each of experts has passed CNCSE".
> > Thanks for pointing out the writting issues.
> >
> > >**Replace the alias tokens in the Open Review abstract with actual terms**
> >
> > Thanks for your comments. We’ll fix it!
> >
> > We really hope our response has addressed your concern. If you have any other questions or suggestions, we are willing to discuss them further.

---

> > > ### Comment · Reviewer_QPXV · 2021-11-28
> > > **Response to the authors**
> > >
> > > Thank you for your responses. I appreciate much content the authors add for addressing the reviewers' concerns.
> > >
> > > >We choose three state-of-the-art and open-sourced models designed for three related benchmarks
> > >
> > > The best performance among them is much lower than I expected. Could the authors provide a discussion to get an improvement by modifying the models? I think Appendix B should be included in the main paper.
> > >
> > > >using NAIL as a pre-training step can significantly improve supervised-learning performance for other tasks
> > >
> > > I'm not sure how we can assess 1.4-2.8 improvements; are they statistically significant? Could the authors provide standard deviations of initial performance (without data augmentation)?
> > >
> > > >How the authors translate Chinese and English examples
> > >
> > > Thank you for the details. In this study, it seems that a lot of experts and native speakers are employed. How did the authors access the pool of those workers, and is the payment scheme fair?
> > >
> > > The other clarifications are useful and helpful. Thank you.
> > >
> > > The authors' responses address some of my comments, but there still remain some concerns. Moreover, I don't have enough time to carefully read through and track all changes of the revised paper and appendices in this limited discussion period, whereas I regret that the authors could put more effort into the first draft. Given these reasons, I cannot change my recommendation score.

---

> > > > ### Author Response · Authors · 2021-11-28
> > > > **Authors Further Response to Reviewer QPXV**
> > > >
> > > > We really thank Reviewer QPXV for checking our reply as well as our revised manuscript!
> > > > We are happy to answer your additional questions.
> > > > >**The best performance among them is much lower than I expected. Could the authors provide a discussion to get an improvement by modifying the models?**
> > > >
> > > > In fact, we have discussed three directions for designing future models in Appendix F.
> > > > + to identify deterministic information or information with low uncertainty.
> > > > + to detect informative subjects/objects.
> > > > + to better utilize elimination.
> > > >
> > > > Since the best performance for sota models on NAIL is low, as for how to adapt these GNN-based models (e.g. DAGN, QDGAT, HGN) to be more suitable for NAIL, specifically, we propose two improvements:
> > > > + One is to focus on the construction of the graph. The graphs should be heterogeneous, including propositional grained, entity grained, candidate answer grained, etc. When designing edges, we should distinguish between affirmative and negative edges, since many statements in the context of NAIL use negative propositions. About attention mechanism, it should be designed to reflect the degree of certainty of information, i.e. nodes that contain more
> > > > deterministic information should be given more weight.
> > > > + The other one is to combine (graph) neural models with a symbolic solver. The neural symbolic network can iteratively judge and verify candidate cases, eliminate the wrong candidate by updating parameters, and finally get the correct answer. We think this is a promising direction and we believe NAIL will contribute to the development of this area.
> > > >
> > > > >**I'm not sure how we can assess 1.4-2.8 improvements; are they statistically significant?**
> > > >
> > > > Thank you for your suggestion. Due to time limitation, we can't do significance checking for all experiments. So far, we have validated the results on two datasets: ReClor and LogiQA. We selected different seeds and conducted several experiments:
> > > >
> > > > | Number | Training Settings |  Evaluation Settings | Results |
> > > > |----|----|----|----|
> > > > | 1 | LogiQA  |  LogiQA  | 35.0±0.3 |
> > > > | 2 | NAIL$\rightarrow$LogiQA | LogiQA  | 36.7±0.2 |
> > > > | 3 | ReClor |ReClor  | 62.4±0.2 |
> > > > | 4 | NAIL$\rightarrow$ReClor | ReClor  | 64.1±0.3|
> > > >
> > > > Using NAIL as extra training resource, the improvements of supervised-learning results on LogiQA and ReClor are statistically significant.
> > > >
> > > >
> > > >
> > > > >**How did the authors access the pool of those workers, and is the payment scheme fair?**
> > > >
> > > > We worked with two outsourcing companies for accessing experts and native speakers respectively.
> > > >
> > > > We ensure that all workers’ privacy rights are respected in the annotation process.
> > > > All workers have been paid above local minimum wage and consented to use the datasets for research purposes covered in our paper.
> > > > Specifically, each expert can receive RMB¥1.5 for answering each example, which generally costs the person 2-3 minutes. Each Chinese/English native speaker can receive RMB¥1.5 / $0.5 for reading through the translation version of a sample, which costs about 1 minute.
> > > >
> > > >
> > > > **Thanks again for your time and encouraging feedback. We hope our response and clarifications have addressed your concerns. We would be grateful if you could re-evaluate our work.**

---

### Official Review · Reviewer_xN3f · 2021-11-01

**Correctness:** 3
**Technical Novelty And Significance:** 1
**Empirical Novelty And Significance:** 1
**Recommendation:** 3
**Confidence:** 5

**Main Review:**

The authors manually constructed a dataset that aims to train and evaluate the model’s capabilities in naïve logical reasoning, which consists of 10,296 instances that were extracted or re-written form Chinese National Civil Servants Examination (CNCSE) and Law School Admission Test (LSAT) by humans.

Strengths:

A dataset was built to train and evaluate model’s capabilities in naïve logical reasoning, which consists of 10,296 instances.

Weaknesses:

All the instances in the dataset were selected or written by humans, and there is no valuable or useful method proposed in the dataset-construction process.
It is not clear how they guarantee the quality of translation. There are even many grammatical errors in the examples given in this paper. For example, “Selecting Prince Charming”, “Xiao Li’s ideal gift has following characters”, etc.
In my opinion, it seems that the main difficulty of answering the questions in the constructed dataset lies in semantic parsing rather than logical reasoning. If the texts are parsed precisely, the models might achieve high accuracy with an appropriate solving (or searching) technique.


**Summary Of The Paper:**

The authors manually constructed a dataset that aims to train and evaluate model’s capabilities in naïve logical reasoning. The instances were extracted from Chinese National Civil Servants Examination (CNCSE) and Law School Admission Test (LSAT) and those instances were re-written by humans to increase the size of the dataset.



**Summary Of The Review:**

The authors manually constructed a dataset that aims to train and evaluate the model’s capabilities in naïve logical reasoning, which consists of 10,296 instances. The instances were extracted from Chinese National Civil Servants Examination (CNCSE) and Law School Admission Test (LSAT) and those instances were re-written by humans to increase the size of the dataset. However, all the instances in the dataset were selected or written by humans, and there is no valuable or useful method proposed in the dataset-construction process. Besides, it is not clear how they guarantee the quality of translation. There are even many grammatical errors in the examples given in this paper. For example, “Selecting Prince Charming”, “Xiao Li’s ideal gift has following characters”, etc. It seems that the main difficulty of answering the questions in the constructed dataset lies in semantic parsing rather than logical reasoning.

---

> ### Author Response · Authors · 2021-11-23
> **Response to Reviewer xN3f**
>
> Thanks for your comments. We briefly respond to a couple of points as follows.
>
> >**There is no valuable or useful method proposed in the dataset-construction process**
>
> We want to clarify: it is **non-trivial** to build a large number of examples that require naive logical reasoning.
> In fact, designing such examples from scratch requires the immense effort from human experts.
> Thus, we propose to imitate the examples of standardized exams and make the following contributions.
> + We provide 3 possible types of imitations, from easy to difficult: S imitations, SO imitations and SPO imitations.
>  To ensure a diverse semantic surface while keeping the underlying logic of the original example,
>  we adopt SPO imitations (See "Imitation Example" of Section 3.2).
> + To control the quality of imitation, we design strict strategies in the process of human imitation to embrace semantic diversity and ensure logic invariance.
>  For example, iterative imitation, double-checking and so on (See "Quality Control" of Section 3.2).
>
>  Overall, we pave an alternate path for future work: using imitation when original collection is small.
>
> >**It is not clear how they guarantee the quality of translation**
>
> Sorry for misunderstanding how to guarantee the quality of translation, we omitted this due to space limitations.
> In fact, after collecting high-quality mono-collections,
> + We first adopted Google Translation to translate Chinese/English collections into another language.
> + Then employed 10 professional bilingual experts in Chinese and English for manual correction. Bilingual experts were asked to pay attention to logic-invariance and faithfulness during translation.
> + Next, to ensure translation quality, we also adopted the post checking strategy. That is, we asked the 20 human experts mentioned above to solve the translated examples. Each translated example was presented to 3 experts randomly. Since human experts excel in solving naive logical reasoning problems, (i.e. achieve 100% accuracy on mono-collections), if any expert made a mistake on a translated sample or pointed out  "UNABLE_TO_ANSWER'', the translated instance is sent back to the bilingual experts for revision.
>
> We added a subsection to the revision of the paper detailing the translation process (see "Translation Quality Control" of Section 3.2).
>
> >**About such translations in the example, "Selecting Prince Charming", "Xiao Li's ideal gift has following characters"**
>
> Thanks for pointing it out, we realized that the translation in the example is typical Chinglish. This is because this original problem was written in Chinese, and we paid more attention to logic-invariance and faithfulness during translation.
> To tackle this limitation, during this period,
> + We asked 50 native speakers to read through all paragraphs of the translation parts in NAIL and mark 0/1 for each, where 1 stands for a translated sample is idiomatic, and 0 otherwise.
> + Then for all samples marked with 0 (about 20%), the bilingual experts and native speakers will work together to polish them and conform to the target language norms.
> Since the translated version of NAIL has been polished, we re-conducted relevant experiments and updated in the revision of the paper. Interestingly, most results remain unchanged if rounded to one decimal place.
>
> Thanks again for your valuable suggestions.
>
> >**The main difficulty lies in semantic parsing rather than logical reasoning**
>
> We believe that semantic parsing might be **ONE** method to solve naive logical reasoning problems.
> It is true that if the texts are parsed precisely, the models might achieve high accuracy with an appropriate solving (or searching) technique.
> However, obtaining a good semantic parser is often **difficult**, and poor parsing results may lead to **error propagation** [1, 2].
> + On the one hand, training a good parser usually requires a large amount of in-domain **labeled data**, which is costly.
> + On the other hand, off-the-shelf semantic parsers are usually trained on out-of-domain data,
> which may cause **domain conflict** problems and lead to poor parsing results.
>
> Therefore, unsupervised semantic parsing and semantic parsing via domain adaptation may be promising directions for the semantic parsing way to logical reasoning.
> In addition, there are **other various methods** targeting naive logical reasoning, such as reasoning over natural language[3,4].
> It can make better use of the knowledge of pre-trained model.
> Overall, how to solve naive logical reasoning in free text remains an open question.
> The proposed NAIL benchmark aims to offer an opportunity for future research in logical reasoning.
>
> [1]Semantic parsing via paraphrasing
>
> [2]Semantic parsing on Freebase from question-answer pairs
>
> [3]Probabilistic Graph Reasoning for Natural Proof Generation
>
> [4]PRover:Proof generation for interpretable reasoning over rules

---

> > ### Author Response · Authors · 2021-11-23
> > **Continue Response to Reviewer xN3f**
> >
> > We really hope our response has addressed your concern. If you have any other questions or suggestions, we are willing to discuss them further.

---

### Official Review · Reviewer_VxU4 · 2021-11-04

**Correctness:** 4
**Technical Novelty And Significance:** 3
**Empirical Novelty And Significance:** 3
**Recommendation:** 8
**Confidence:** 4

**Main Review:**

A really interesting work. A newly introduced and challenging datasets which will be happily received by the scientific community.
The paper is well organized and written. All the decisions taken by the authors are properly described and justified.
My only concern on a paper of this quality is the lack of experimentation using NAIL as a pre training step for other logical reasoning datasets. If learning the logic of this dataset bring naive reasoning still into a model, that should be reflected into the performance of other logical datasets like LogiQA, ReClor, RACE, but also in CLUTRR, MAthQA, AQuA, HotpotQA among others ...


**Summary Of The Paper:**

This work introduces NAIL, a bilingual (English and Chinese) a benchmark for naive logical reasoning, inspired in the kind of questions, involving this aspect contained in standardized exams such as Chinese
National Civil Servants Examination (CNCSE) and Law School Admission Test (LSAT).
Two different sets are collected in this work. NAIL-E, which are actual examples from both CNCSE and LSAT, translated to English and Chinese respectively, and NAIL-I, a data augmentation approach that uses the structure from existing examples and replaces the three aspects of the problem, subject, predicated and objects in order to create new instances.
An qualitative analysis is carried out to reach the conclusion that those three replacings are necessary in order to create new, and more challenging instances.
The resulting datasets NAIL-E and NAIL-I are finally splitted into train, dev and test, and examined to test their quality.
Random baseline, word match, sliding window, BERT and Roberta (as well as human evaluations carried out by average and expert annotators) are carried out showing the quality of the resulting datasets and the challeding there are for current NLP models.
Extra experimentation is done to test their transfer learning skills among the full dataset, its split NAIL-E and NAIL-I and using LogiQA, ReClor, RACE as pretriaining steps.


**Summary Of The Review:**

Please provide a short summary justifying your recommendation of the paper.
Good paper
Very useful dataset
All reasoning steps in the development are properly justified
Experimentation is reasonable and accurate

---

> ### Author Response · Authors · 2021-11-23
> **Response to Reviewer VxU4**
>
> Thanks for your valuable comments. We briefly respond to a couple of points as follows.
>
> >**Lack of experimentation using NAIL as a pre training step for other logical reasoning datasets**
>
> Thanks for your kind advice.
> We conduct such experiments supplementally. We use NAIL as extra training resource for RoBERTa for learning existing reading comprehension tasks, where LogiQA, ReClor, RACE, HotpotQA, MathQA, AQuA are adopted (experiments on CLUTRR, AQuA are ongoing, and the results will be updated in time), and supervised-learning results are listed as follows.
>
> |Number| Training Settings |  Evaluation Settings |Results|
> |----|----|----|----|
> | 1 | LogiQA  |  LogiQA  | 35.3 |
> | 2 | NAIL$\rightarrow$LogiQA | LogiQA  | 36.9 (+1.6)|
> | 3 | ReClor |ReClor  | 62.6 |
> | 4 | NAIL$\rightarrow$ReClor | ReClor  | 64.4 (+1.8)|
> | 5 | RACE  |RACE | 83.2 |
> | 6 | NAIL$\rightarrow$RACE |RACE | 85.2 (+2.0)|
> | 7 | MathQA | MathQA    | 39.8  |
> | 8 | NAIL $\rightarrow$ MathQA |  MathQA | 41.2 (+1.4)|
> | 9 | HotpotQA|HotpotQA    | 69.8  |
> | 10 | NAIL $\rightarrow$ HotpotQA| HotpotQA | 72.6 (+2.8)|
>
> Generally, we can draw the conclusion that using NAIL as a pre-training step can significantly improve supervised-learning performance for other tasks, with boosts ranging from 1.4 points to 2.8 points. This indicates that NAIL can bring naive logical reasoning ability to the model, which is a basic reasoning ability and can be reflected into other comprehension tasks, such as solving math word problems (MathQA), multi-hop factual reasoning (HotpotQA), various logical reasoning (LogiQA and ReClor), and general understanding (RACE).
> More details can be found in "Transfer Learning" in Section 4 and Appendix D.2.
>
> We also conduct zero-shot experiments, that is, using NAIL as training resource, and test directly on the mentioned datasets (LogiQA, ReClor, and RACE).
> For more details, please refer to Figure 4 in "Transfer Learning" in Section 4 and Appendix D.2.
> For more error analysis,  please refer to  "Error Analysis" in Section 4 in the revised paper.
>
> Thanks again for your valuable suggestions.

---

> > ### Author Response · Authors · 2021-11-23
> > **Continue Response to Reviewer VxU4**
> >
> > We really hope our response has addressed your concern. If you have any other questions or suggestions, we are willing to discuss them further.

---

### Official Review · Reviewer_kWbB · 2021-11-05

**Correctness:** 3
**Technical Novelty And Significance:** 3
**Empirical Novelty And Significance:** 2
**Recommendation:** 6
**Confidence:** 4

**Main Review:**

Naive logical reasoning (NAIL) is quite difficult even for humans: even well trained human experts cannot agree on all the answers (such ambiguous answers were left out from the final dataset), and human non-experts score much lower than experts. Therefore, it is expected that the task will be extremely difficult for language models, and the gap between human and system performance is indeed prominent, and calls for future developments in improving logical ability of large neural models.

As said, the main contributions of the paper are: 1) creation of the dataset for the NAIL task, 2) showing that some pretrained language models are not as good as humans in solving this task. However, I believe that the paper still does not paint the whole picture and should bring much more detailed analyses on the current gaps, why they happen, and how we can address those gaps.

I am not fully convinced about motivation to have this sort of logical reasoning embedded in the model - these NAIL tasks seem more like 'enigmatic' puzzles, and a stronger motivation is needed to justify why pretrained language models should even need the ability to solve such puzzles: can they be fully trained (with larger datasets) to do well on the puzzles? Another question is, where I expected a deeper discussion: do we need AGI to solve puzzles like the one presented in NAIL? Is it a data problem or truly a human-like reasoning problem? In other words, with even larger models (GPT-3, T5) and more training data, can we expect the models to close this gap to human (non-expert) performance?

Overall, the paper is quite thin on additional insightful experiments and further analyses, going beyond reporting the dataset creation protocol and benchmarking a subset of models on the data, and does not read as a complete piece of work.

While a direct link is not apparent at first sight, this work also reminds me very much of the ACL 2020 work of Sahin et al. (https://aclanthology.org/2020.acl-main.115/) - they also showed how pretrained models do lack the skill of iterative reasoning upon knowledge. In other words, supported by a slightly different iterative logical reasoning task than NAIL, they reach exactly the same conclusion. One could read that this lack of ability has been shown in the previous paper, and this work just restates with another similar task, which also diminish novelty of this work a bit.

Given that imitation examples rely on the same backbone template, and the only variation is actually in the semantics of the examples (i.e., variation is at word-level or 'discourse'-level), I would like to understand more why such imitation examples do help increase the scores: is it only because of a higher semantic coverage, or is it about some spurious artefacts that the model learns by seeing more examples of the same backbone type (e.g., some hidden correlation learned by the model). Would it be possible to create imitation examples automatically and augment training data with such examples? How would the models then behave?

Only performance of BERT and RoBERTa is attested in the task - I would very much suggest to try out other language models (e.g., T5) and think of possible (auxiliary/intermediate) tasks which could prepare off-the-shelf models to become better at this type of task.

**Summary Of The Paper:**

This paper proposes a new challenge for large pretrained models (which can of course extend to any other model claiming to store language knowledge): naive logical reasoning. The main contribution of the paper is the construction of a dataset in Chinese and English (termed NAIL) with items for training and evaluating such naive logical reasoning, where the task in the dataset is cast as a standard multi-choice (4-choice in this case) question, with only one correct answer. The main empirical result of the paper consists in showing that pretrained language models, even when trained on naive logical reasoning examples, lag far behind human performance.

**Summary Of The Review:**

Similar to some prior work, the paper exposes the inability of (a subset of) pretrained language models (BERT, RoBERTa) to do complex logical reasoning, showcasing that their performance is much lower than what humans score. This is done through the introduction of a new task called naive logical reasoning. Overall, while the NAIL dataset can be quite helpful in guiding future advances on equipping language models with knowledge and methods required to solve such logical 'puzzles', the paper should have done a much better work on the aspects of motivation, discussion, analyses, and higher-level implications of the main results. I feel that more work is needed (see the main review).

---

> ### Author Response · Authors · 2021-11-23
> **Response to Reviewer kWbB**
>
> Thanks for your review comments. We briefly respond to your questions as follows.
>
> >**About motivation to have this sort of logical reasoning embedded in the model**
>
> We hope that the proposed NAIL inspires further efforts towards human-like logical reasoning in NLP.
> To move towards human intelligence, we need to equip the models with logical reasoning capabilities (e.g.,  ability to draw logical conclusions from given statements).
> And this is also a long sought-after goal of AI.
>
> We would like to clarify that the pre-trained model is **NOT** the only solution.
> The current consensus is that the pre-trained models have only shallow reasoning capabilities and experiments show that  pre-trained models perform poorly on this task.
> While traditional symbolic reasoning is better suitable for this task, but it also suffers from the difficulty of semantic parsing.
> Thus how to combine traditional symbolic reasoning and neural networks ( denoted as neural-symbolic learning) is a promising direction. We believe NAIL will contribute to the development of this area.
>
>
> >**Do we need AGI to solve puzzles like the one presented in NAIL?  Is it a data problem or truly a human-like reasoning problem? In other words, with even larger models (GPT-3, T5) and more training data, can we expect the models to close this gap to human (non-expert) performance?**
>
> We think it is a human-like reasoning problem.
> It is predictable that even with more powerful pre-trained models (e.g., T5), it will fall far short of solving this type of problem.
> It would be better performance if there are more training data.
> However, it is impractical to build such a large-scale dataset from scratch.
> So it is not known how much data is required to reach human level.
> In contrast to humans, this reasoning capability can be learned from few examples.
> In practice, we prefer to learn reasoning capability with a moderate amount of data.
> Therefore, we propose the NAIL to evaluate the logical reasoning capability.
> Also, we encourage various methods to solve this dataset, including symbolic reasoning, pre-trained model, and AGI.
>
>
> >**Additional insightful experiments and further analyses**
>
> Thanks for your suggestions.
> We have omitted some parts of the experiments and further analyses in the draft version due to space limitations.
> In the updated version, we have included more experiments and analyses.
> See Section 4,
> + in the "Transfer Learning" paragraph, including using NAIL/other datasets as extra training resources, conducting zero-shot and supervised-setting experiments on other datasets/NAIL, respectively.
> + in the "Error Analysis" paragraph, we measure human-baseline errors and model errors against several factors: number of sentences in the backbone template of the context , number of possible worlds, and context length.
> See Appendix B, we add more strong and complicated baselines involving generation-based methods and well-designed models for other related tasks.
> See Appendix F, we add more discussions about future directions for models to solve NAIL.
>
> >**About the ACL2020 work of Sahin et al**
>
> Current deep learning is most successful at perception tasks and generally what are called **system 1 tasks**.
> Using deep learning for **system 2 tasks** that require a deliberate sequence of steps is an exciting area, which is still in its infancy.
> Our work and the ACL2020 work of Sahin et al. both focus on System 2.
> However, the reasoning capability in these two works are very different.
> Our work focuses on a fine-grained type of logical reasoning, while the other work focuses on meta-linguistic reasoning.
> The solutions for these two reasoning capability may be very different and deserve to be studied separately.
>
>
> >**Why such imitation examples do help increase scores**
>
> The performance of most deep models depends on the quantity and diversity of data.
> However, the number of original collections in NAIL-E is too small (only 488 examples).
> NAIL expanded NAIL-E by human imitation, thus reducing data overfitting and introducing variability to data.
> This is similar to the effect of data augmentation. However, because problems belonging to naive logical reasoning category are hard to augment automatically (as stated in the next question), we bring human augmentation into the collection of such problems. Since creating such problems requires immense efforts from human experts, we propose the human imitation method.

---

> > ### Author Response · Authors · 2021-11-23
> > **Continue Response to Reviewer kWbB**
> >
> > >**Would it be possible to create imitation examples automatically and augment training data with such examples? How would the models then behave?**
> >
> > In fact, automatically creating more data can lead to imperfect imitations of low quality(e.g., S imitation and SO imitation). It can only create superficially very similar examples in a limited  way.
> > We think that it may improve the robustness of the model but does not easily lead to large improvements.
> >
> > For SPO imitation, it is difficult to automatically create examples.
> > The reason is that an instance contains multiple predicates with constraints between them, and replacing one of them at random would result in an obvious mismatch.
> > We will include it and make the analyses in the appendix.
> > Thanks for your advice.
> >
> > >**Only performance of BERT and RoBERTa is attested in the task**
> >
> > We include T5 in the experiment supplementally.
> > Due to computational resource limitation, we only try an off-the-shelf T5-base model.
> > Results are shown below, showing that T5 is a very powerful generative model, but it seems that T5 lacks naive reasoning capability as well.
> >
> > | MODEL | DEV | TEST |
> > |----|----|----|
> > | T5-base | 27.1 | 26.3 |
> > | RoBERTa-base | 27.6 | 26.4 |
> > | RoBERTa-large | 34.6 | 30.1 |
> >
> >
> > We also choose three state-of-the-art and open-sourced models designed for three benchmarks: DAGN for LogiQA and Reclor, HGN for HotpotQA, and QDGAT for DROP respectively. We reproduced these three models with the code released by authors and NAIL as training data:
> >
> > | MODEL | ENCODER|DEV| TEST|
> > |----|----|----|----|
> > | RoBERTa-large | RoBERTa-large | 34.6 | 30.1 |
> > | DAGN | RoBERTa-large | 36.0 | 32.3 |
> > | HGN | RoBERTa-large | 30.1 | 28.7 |
> > | QDGAT | RoBERTa-large | 28.3 | 28.7 |
> >
> > We observe that DAGN achieves the best results on NAIL, since ReClor and LogiQA are much closer to NAIL, the well-designed discourse-aware graph network DAGN is more applicable to NAIL.
> > However, there is still very much space for models' improvement compared to humans.
> > For more details, please refer to Appendix B in the revised paper.
> >
> > Thank you for the suggestion.
> >
> > We really hope our response has addressed your concern. If you have any other questions or suggestions, we are willing to discuss them further.

---

> > > ### Comment · Reviewer_kWbB · 2021-11-29
> > > **Thanks for the detailed response!**
> > >
> > > I would like to thank the authors for going the extra mile with their responses and additional experiments conducted. Most of my questions were addressed to a very satisfying manner and I'm happy to raise my score.

---

### Author Response · Authors · 2021-11-23
**General Response to all reviewers**

We sincerely thank all reviewers for their insightful comments. We have uploaded a new version of our draft, adding more experiments and analyses. The main modifications are as follows:
+ In the "Translation Quality Control" paragraph in Section 3.2, we described details about the translation process.
+  In the "Transfer Learning" paragraph in Section 4, we add more experiments: using NAIL/other datasets as extra training resources, conducting zero-shot and supervised-setting experiments on other datasets/NAIL, respectively.
+ In the "Error Analysis" paragraph in Section 4, we measure human-baseline errors and model errors against several factors: number of sentences in the backbone template of the context , number of possible worlds, and context length.
+ In Appendix B, we add more strong and complicated baselines involving generation-based methods and well-designed graph-based models for other related tasks.
+ In Appendix F, we add more discussions about future directions for models to solve NAIL.
+ Experiments about the effect of NAIL, NAIL-E and NAIL-I are moved to Appendix C.

---

### Author Response · Authors · 2021-11-27
**Looking forward to further feedbacks**

Dear ACs and Reviewers,

Thank you again for your great efforts and the valuable comments. We have responded carefully and in detail to the main concerns. We hope you will be satisfied with these responses.

As the discussion phase is about to close I.e. 29th, we are very much looking forward to hearing from you about any further feedback. We will be delighted to clarify any further concerns (if any).

Thank you very much for reading this letter in such a busy schedule.

Best,
Authors

---

### Decision · Program_Chairs · 2022-01-20

**Decision:**

Reject

**Comment:**

This paper introduces a dataset, based on preexisting standardized tests, of elimination/grid-completion-style logical reasoning puzzles expressed in text; available in both Chinese and English (with some of the text coming from semi-automatic translation). The early pretrained MLMs BERT and RoBERTa perform poorly.

This paper is solidly borderline. Reviewers had some concerns about the motivation and novelty the work, but I think that there is a plausible enough story for where this data will have value that I'm not comfortable rejecting it only on this basis. More worryingly, the initial submission had some fairly serious writing quality and clarity issues, which impacted both the paper *and* the data. It seems like the authors made significant progress on this in the revision and engaged substantially during the discussion, but reviewers were not fully satisfied that the paper was up to ICLR standards, either as an initial submission or after revisions. This is a small detail, but it's a bad sign for the carefulness of the work that the OpenReview abstract is still unreadable, even after a request from a reviewer.